# Baseline Absolute Lymphocyte Count and ECOG Performance Score Are Associated with Survival in Advanced Non-Small Cell Lung Cancer Undergoing PD-1/PD-L1 Blockade

**DOI:** 10.3390/jcm8071014

**Published:** 2019-07-10

**Authors:** Florian Huemer, David Lang, Theresa Westphal, Simon Peter Gampenrieder, Georg Hutarew, Lukas Weiss, Hubert Hackl, Bernd Lamprecht, Gabriel Rinnerthaler, Richard Greil

**Affiliations:** 1Department of Internal Medicine III with Haematology, Medical Oncology, Haemostaseology, Infectiology and Rheumatology, Oncologic Center, Salzburg Cancer Research Institute-Laboratory for Immunological and Molecular Cancer Research (SCRI-LIMCR), Paracelsus Medical University, 5020 Salzburg, Austria; 2Department of Pulmonology, Kepler University Hospital, Med Campus III, 4020 Linz, Austria; 3Institute of Pathology, Paracelsus Medical University Salzburg, 5020 Salzburg, Austria; 4Division of Bioinformatics, Biocenter, Medical University of Innsbruck, 6020 Innsbruck, Austria

**Keywords:** absolute lymphocyte count, ECOG performance status, immune-checkpoint inhibitor, antibiotics, PD-1/PD-L1, immune-checkpoint blockade, RANK, VEGF, denosumab

## Abstract

Immune-checkpoint blockade in front-line or second-line treatment improves survival in advanced non-small cell lung cancer (aNSCLC) when compared with chemotherapy alone. However, easily applicable predictive parameters are necessary to guide immune-checkpoint inhibition in clinical practice. In this retrospective bi-centric analysis, we investigated the impact of baseline patient and tumor characteristics on clinical outcome in aNSCLC patients treated with programmed cell death protein 1(PD-1)/programmed cell death ligand 1 (PD-L1) inhibitors. Between May 2015 and January 2018, 142 unselected consecutive NSCLC patients received PD-1/PD-L1 inhibitors during the course of disease. In multivariate analysis, we identified the Eastern Cooperative Oncology Group (ECOG) performance status (ECOG > 1 versus ECOG ≤ 1, HR: 3.23, 95%CI: 1.58–6.60, *P* = 0.001), baseline absolute lymphocyte count (ALC; high: >0.93 × 10^9^/L versus low: ≤ 0.93 × 10^9^/L, HR: 0.38, 95%CI: 0.23–0.62, *P* < 0.001), prior or concomitant anti-vascular endothelial growth factor (VEGF) targeting therapy (yes versus no, HR: 2.18, 95%CI: 1.15–4.14, *P* = 0.017) and TNM stage (IV versus III, HR: 4.18, 95%CI: 1.01–17.36, *P* = 0.049) as the most relevant parameters for survival. Neither antibiotic exposure (antibiotic-positive versus antibiotic-negative, HR: 0.90, 95%CI: 0.56–1.45, *P* = 0.675), nor PD-L1 expression on tumor cells (≥1% versus <1%, HR: 0.68, 95%CI: 0.41–1.13, *P* = 0.140) was associated with survival. Baseline ECOG performance status and ALC were associated with survival in aNSCLC patients treated with PD-1/PD-L1 inhibitors and assessment of these parameters could be suitable in clinical practice.

## 1. Introduction

Non-small cell lung cancer (NSCLC) represents the leading cause of cancer-mortality in the United States and in Europe [1,2]. The therapeutic concept of unleashing a pre-existing immune response against the tumor by the use of immune-checkpoint inhibitors results in long-term survival in 17% to 27% of patients with advanced NSCLC (aNSCLC) [3,4]. Meanwhile, positive phase III trial data support the use of nivolumab [5,6], pembrolizumab [7,8,9,10], atezolizumab [11,12], durvalumab [13], and ipilimumab combined with nivolumab [14] for systemic therapy in aNSCLC.

Despite achievement of overall response rates (ORR) between 45% to 48% by immune-checkpoint blockade (ICB) as monotherapy in programmed cell death ligand 1 (PD-L1) high expressing tumors or in combination with chemotherapy and despite a plateauing of overall survival (OS) curves after 12 to 15 months, half of the patients experience disease progression after 8.8 to 10.3 months from initiation of first-line therapy and will necessitate further systemic treatment [8,9]. Due to the rising costs caused by the therapeutic approach of ICB and, in order to select patients that will derive clinical benefit from currently approved immune-checkpoint inhibitor protocols, the identification of easily available predictive baseline parameters is an absolute necessity in the daily clinical practice. Several attempts have been made to predict the therapy response or treatment failure to ICB based on patient and tumor characteristics.

The PD-L1 expression on tumor cells has been extensively investigated as a predictive marker for ICB. Although the clinical outcome on immune-checkpoint inhibition in aNSCLC improves with higher PD-L1 expression [6,7,9,10,11,12,15], PD-L1 as a biomarker has several limitations. PD-L1 expression is heterogeneous and influenced by chemotherapy and targeted therapy [16]. Furthermore, staining differences among various antibody clones pose a challenge in clinical practice and PD-L1 negativity does not exclude a response to immune-checkpoint inhibition [17].

Results of a few recently published retrospective studies demonstrate a negative impact of antibiotic use in temporal proximity to the initiation of ICB for several tumor entities including NSCLC [18,19,20,21,22]. Fecal microbiota transplantation from cancer patients who responded to immune-checkpoint inhibitors into mice that had been pretreated with antibiotics restored responses to ICB in these animals [18]. The latter finding suggests a substantial role of the gut microbiota composition during immune-checkpoint inhibitor therapy.

Tumor mutational burden (TMB) represents another biomarker of interest to guide immune-checkpoint inhibitor therapy in various tumor entities [23]. Presentation of neoantigens generated by tumor somatic mutations is essential for tumor immunogenicity and the response to ICB [24,25], whereas a TMB cut-off of ≥10 mutations per mega base was clinically validated in NSCLC [14,26].

Already established biomarkers such as driver mutations of the epidermal growth factor receptor (EGFR), which are generally sensitive to tyrosine-kinase inhibitors, also play a role when considering ICB since rising evidence suggests that single-agent immune-checkpoint inhibition is not active in NSCLC with sensitizing EGFR mutations [6,27,28].

Inflammation is involved in the pathogenesis and promotion of cancer progression [29,30] and is associated with worse clinical outcome in aNSCLC [31]. The suppression of lymphocyte and natural killer cell activity, the release of cytokines, which support tumor progression, and an adverse influence on the tumor microenvironment may contribute to a worse clinical outcome in a state of chronic inflammation [32]. Besides the C-reactive protein, the neutrophil-lymphocyte ratio (NLR) represents a marker of systemic inflammation, is of prognostic value in many solid tumors including NSCLC [32,33], and has also been proposed to predict the benefit from immune-checkpoint inhibition in aNSCLC [34,35,36,37].

Anti-vascular endothelial growth factor (VEGF) targeting therapy with the monoclonal antibody bevacizumab has been used in combination with chemotherapy in aNSCLC with non-squamous histology [38] and is recommended as front-line therapy in combination with chemotherapy and PD-L1 inhibition by the current NCCN guidelines in non-squamous histology aNSCLC [39]. Apart from inhibiting tumor angiogenesis, anti-VEGF targeting therapy exerts immunomodulatory effects [40] and, therefore, is an interesting combination partner for ICB.

Development of bone metastases during the course of aNSCLC is frequently observed. The monoclonal antibody denosumab inhibits osteoclast maturation [41], decreases skeletal-related events in patients with bone metastases from solid tumors [42], improves survival in aNSCLC with bone metastases [43], and is approved for the latter indication. Furthermore, denosumab acts as an immunomodulator [44,45], but its influence on ICB activity in vivo has to be clarified.

In this retrospective bicentric study, we report the impact of baseline patient characteristics, baseline laboratory parameters, and tumor characteristics on clinical outcome with programmed cell death protein 1 (PD-1)/PD-L1 inhibitors in a well-characterized advanced NSCLC cohort.

## 2. Patients and Methods

### 2.1. Patients

In this retrospective analysis, we included unselected consecutive patients with histologically confirmed aNSCLC (stage III/IV) that had been treated with immune-checkpoint inhibitors at the tertiary cancer centers in Salzburg (Austria) and Linz (Austria). Upon first presentation of positive phase III trial data for second-line aNSCLC [5,6,7] PD-1/PD-L1 inhibitors were applied as monotherapy within named patient programs, before the respective approval by the European Medicines Agency (EMA) and Food and Drug Administration (FDA). After drug approval and incorporation of immune-checkpoint inhibitors into the guidelines of the European Society of Medical Oncology (ESMO) and National Comprehensive Cancer Network (NCCN), these guidelines were followed. Patients receiving PD-1/PD-L1 inhibitors in combination with chemotherapy and anti-VEGF targeting therapy within clinical trials (clinicaltrials.gov identifier NCT02367794 and NCT02367794) were also included in this analysis.

### 2.2. Data Collection

Baseline patient characteristics and baseline laboratory values preceding the initiation of immune-checkpoint inhibitor therapy up to 14 days were retrospectively assessed. Data were extracted from medical records including age, sex, Eastern Cooperative Oncology Group (ECOG) performance status, TNM stage, histologic subtype, smoking history, EGFR mutation status, anaplastic lymphoma kinase (ALK) translocation status, central nervous systems (CNS) involvement, PD-L1 expression on tumor cells, immune-checkpoint inhibitor therapy line, immune-checkpoint inhibitor substance, prior or concomitant denosumab application, prior or concomitant anti-VEGF targeting therapy (bevacizumab, ramucirumab, and nintedanib), antibiotic treatment status, subsequent therapy protocols, absolute lymphocyte count (ALC), absolute neutrophil count (ANC), NLR (calculated as the ratio of the baseline ANC and ALC), C-reactive protein (CRP), and prior radiotherapy to the primary tumor or metastases.

Central assessment of the PD-L1 expression status for both oncologic centers was carried out by a single experienced lung pathologist. PD-L1 expression on tumor cells was assessed by immunohistochemistry utilizing the anti-PD-L1 clone 22C3 from Dako^®^. In immune-checkpoint inhibitor clinical trials in aNSCLC a PD-L1 (tumor propensity score) cut-off value of ≥1% is frequently used for stratification and defined PD-L1 positivity in our analysis. Due to the low frequency of EGFR driver mutations among squamous NSCLC, testing for these oncogenic aberrations is not routinely carried out in our clinical practice and, therefore, these cases were considered as EGFR wild-type [46].

Radiologic reassessment by PET-CT or CT scan was performed every two to three months, or as clinically indicated. Progression-free survival (PFS) was calculated from the date of start of ICB until radiologically confirmed progression or death. Patients without progression at the last contact were censored. OS was calculated from the date of ICB initiation until death from any cause. Patients alive at the last contact were censored. Concomitant use of antibiotics was defined as the application within a time frame of one month before or one month after the initiation of ICB.

### 2.3. Statistical Analyses

Differences in patient baseline characteristics between anti-VEGF therapy exposed and anti-VEGF naïve patients were tested by Pearson’s χ^2^-test. For continuous data, the difference between the two groups was calculated with two-sided Wilcoxon rank-sum test. Maximal Harrell’s C-index was used to find the optimal cut-off value for OS prediction for continuous data such as the ALC, ANC, and NLR (Appendix A). In an exploratory analysis, we used the Kaplan–Meier method for survival curves and to evaluate PFS and OS differences according to baseline characteristics. Log-rank test was used to compare survival distributions between two patient groups. Median follow-up time was calculated using Kaplan-Meier curves where event indices (death versus censor) were switched. Kendall’s tau coefficient (Kendall tau-b) was used to measure ordinal associations between parameters. Univariate Cox regression analyses were performed on OS and PFS for indicated, dichotomized, or binary patient data. Only significant variables in the univariate test (*P* < 0.05, Wald test) were included in multivariate Cox regression models with an exception for NLR since it is directly dependent on ALC. We performed additional multivariate Cox regression analyses including clinically important parameters independent of significance (histology, age, and sex) and including all variables besides NLR and ANC. Proportional hazard assumptions were tested using the ‘coxzph’ function. All analyses were performed using the statistical software environment R (version 3.5.1, www.R-project.org, Vienna, Austria) including package ‘survival.’

## 3. Results

Between May 2015 and January 2018, 142 patients with aNSCLC were treated with PD-1/PD-L1 inhibitors at the two tertiary cancer centers. At data cut-off (10 January 2018) after a median follow-up of 13.3 months, 109 patients had progressed on immune-checkpoint inhibitor therapy and 76 patients had died. The baseline characteristics are depicted in Table 1.

### 3.1. Progression-Free Survival

Median PFS was 3.9 months (95%CI: 3.1–5.7 months, Appendix A).

In univariate analysis, the TNM stage (IV versus III, HR: 2.03, 95%CI: 1.02–4.03, *P* = 0.044), immune-checkpoint inhibitor therapy line (≥3rd line versus <3rd line, HR: 1.73, 95%CI: 1.13–2.63, *P* = 0.011), PD-L1 expression status (positive versus negative, HR: 0.49, 95%CI: 0.32–0.75, *P* = 0.001, Appendix A) (different cut-offs for PD-L1 (<1%, 1–50%, >50%) resulted in a significant association with PFS, logrank *P* = 0.0038), prior or concomitant application of anti-VEGF targeting therapy (yes versus no, HR: 1.89, 95%CI: 1.07–3.36, *P* = 0.029), ECOG performance status (>1 versus ≤1, HR: 2.53, 95%CI: 1.49–4.29, *P* < 0.001) and ALC (high: >0.93 × 10^9^/L versus low: ≤0.93 × 10^9^/L, HR: 0.58, 95%CI: 0.38–0.88, *P* = 0.010) were statistically significantly associated with PFS. In multivariate analysis, only baseline ALC remained independently associated with PFS (HR: 0.55, 95% CI: 0.35–0.88, *P* = 0.012, Table 2A). ALC were still significantly associated with PFS for both if histology, age, and sex were included in the multivariate model (*P* = 0.030) and if all other variables besides NLR and ANC were included in a multivariate model (*P* = 0.023) (Table 2A).

### 3.2. Overall Survival

Median OS was 12.2 months (95% CI: 10.7–15.1 months, Appendix A). In univariate analysis, baseline NLR (high: >3.8 versus low: ≤3.8, HR: 2.22, 95%CI: 1.36–3.63, *P* = 0.001, Figure 1A), TNM stage (IV versus III, HR: 5.52, 95% CI: 1.35–22.64, *P* = 0.018, Figure 2A), immune-checkpoint inhibitor therapy line (≥3rd line versus <3rd line, HR: 1.97, 95%CI: 1.24–3.16, *P* = 0.004), prior or concomitant application of anti-VEGF targeting therapy (yes versus no, HR: 2.18, 95% CI: 1.21–3.93, *P* = 0.010, Figure 2B), ECOG performance status (ECOG > 1 versus ECOG ≤ 1, HR: 2.58, 95% CI: 1.30–5.10, *P* = 0.007, Figure 2C), and ALC (high: >0.93 × 10^9^/L versus low: ≤0.93 × 10^9^/L, HR: 0.38, 95% CI: 0.24–0.62, *P* < 0.001, Figure 1B) showed a statistically significant association with OS (Table 2B). Due to the fact that differences in OS according to the NLR (Figure 1A) were mainly attributable to variations in the ALC (Figure 1B), but not to variations in the ANC (Figure 1C), only the ALC were tested in multivariate analysis. In multivariate analysis, the TNM stage (HR: 4.18, 95%CI: 1.01–17.4, *P* = 0.049,), prior or concomitant anti-VEGF targeting therapy (HR: 2.18, 95%CI: 1.15–4.14, *P* = 0.017), ECOG performance status (HR: 3.23, 95%CI: 1.58–6.60, *P* = 0.001), and ALC (HR: 0.38, 95%CI: 0.23–0.62, *P* < 0.001), remained statistically significantly associated with OS (Table 2B). ALC were also significantly associated with OS for both, if additionally histology, age, and sex, were included in the multivariate model (*P* < 0.001) and if all other variables besides NLR and ANC were included in a multivariate model (*P* = 0.003) (Table 2B).

ALC and ECOG performance status remained significant in overall survival estimates adjusted for the therapy-line (1 + 2 versus ≥ 3; Appendix A).

Contrary to our hypothesis that anti-VEGF targeting therapy may improve the clinical outcome with PD-1/PD-L1 inhibitors, prior or concomitant anti-VEGF targeting therapy was associated with a significantly inferior survival. In order to disclose differences that may explain this unexpected finding, baseline characteristics between anti-VEGF naïve and anti-VEGF exposed patients were compared. Substantial differences concerning PD-L1 positivity (67% versus 36%, *P* = 0.024) and concerning administration of ICB as the third line therapy and beyond (20% versus 59%, *P* = 0.002) were found (Appendix A).

### 3.3. Association of ALC, ECOG Performance Status and PD-L1 Expression

The baseline ALC and baseline ECOG performance status showed a statistically significant weak inverse correlation (Kendall tau-b = −0.17, *P* = 0.010, Figure 3A). Categorical PD-L1 expression was neither associated with the ECOG performance status (Kendall tau-b = −0.088, *P* = 0.298, Figure 3B) nor with the ALC (Kendal tau-b = −0.007, *P* = 0.923, Figure 3C).

Concomitant application of antibiotics with initiation of PD-1/PD-L1 blockade did neither affect median PFS (AB^+^: 3.8 months (95%CI: 2.5–7.4) versus AB^−^: 4.0 months (95%CI: 3.1–5.7), HR: 1.02 (95%CI: 0.69–1.50), *P* = 0.920, Appendix A) nor median OS (AB^+^: 14.6 months (95%CI: 9.4–NA) versus AB^−^: 11.2 months (95%CI: 9.9–15.1), HR: 0.91 (95%CI: 0.57–1.45), *P* = 0.675, Appendix AB) in the entire cohort.

## 4. Discussion

In this retrospective analysis, the TNM stage (IV versus III, HR 4.18), prior anti-VEGF targeting therapy (yes versus no, HR: 2.18), ECOG performance status (ECOG > 1 versus ECOG ≤ 1, HR: 3.23), and ALC (high: >0.93 × 10^9^/L versus ≤0.93 × 10^9^/L, HR: 0.38) were identified as the most relevant baseline parameters associated with OS in aNSCLC treated with ICB (Table 2B). We are convinced that assessment of baseline ECOG performance status and ALC before the initiation of ICB is routinely carried out and, thus, feasible for clinical use in any lung cancer unit.

Subgroup analyses of the vast majority of trials showed that a better baseline ECOG performance status is associated with a superior clinical outcome in patients with aNSCLC undergoing palliative immune-checkpoint inhibitor therapy [5,6,8,9,10,12], chemotherapy [47,48,49], or targeted therapy [50,51]. Considering the impact of the ECOG performance status on OS irrespective of the type of systemic therapy, this parameter is more likely to be of a prognostic value than of a predictive value. Due to the fact that, in the general inclusion of NSCLC patients in immune-checkpoint inhibitor trials is restricted to an ECOG performance status ≤1 [5,8,9,10,12], the actual impact of the ECOG performance status on OS is not depicted in these studies.

As a surrogate marker for systemic inflammation, the baseline NLR at initiation of ICB was significantly associated with OS in our NSCLC cohort. In general, the favorable outcome of ICB in patients with a low baseline NLR does not come at the cost of increased toxicity [52]. Due to the retrospective nature of this study, assessment of immune-related adverse events was not performed. However, in our NSCLC cohort, the impact of the NLR on OS (Figure 1A) was predominantly driven by variations in the ALC (Figure 1B), while variations in the ANC played a subsidiary role (Figure 1C). The calculated ALC cut-off in our study closely resembles the lower limit of normal and is, therefore, suitable for use in clinical practice. Baseline CRP levels above the upper limit of normal (>0.6 mg/dL) did neither affect PFS nor OS (Table 2), which was also the case for higher CRP cut-off values (>5 mg/dL versus ≤5 mg/dL). Although ECOG performance status and ALC proved to be independently associated with OS in multivariate analysis, a highly significant inverse correlation was found between these two parameters (Figure 3A). The latter finding is in line with previous reports [53].

In contrast to the majority of phase III trial data [6,7,8,9,10,11,12], in our NSCLC cohort, the PD-L1 expression on tumor cells was not associated with OS (Appendix A). It is noteworthy that 73% of patients received ICB as second or later-line therapy in our cohort. Due to prior systemic therapy application and due to the fact that PD-L1 expression is a dynamic parameter, the PD-L1 expression reported in the initial diagnostic biopsy may not reflect the actual PD-L1 expression at the time point of ICB initiation [16]. Central assessment of the PD-L1 expression status for both oncologic centers was carried out by a single experienced lung pathologist and, therefore, staining results could not be biased by inter-observer variations.

Antibiotic exposure in temporal proximity to the initiation of ICB did neither impact PFS (Appendix A) nor OS (Appendix A) in our bicentric NSCLC cohort. Published data on the influence of antibiotics on clinical outcome are conflicting [54]. While several retrospective studies reported a detrimental effect of antibiotic exposure in temporal proximity to the start of ICB in aNSCLC [18,20,22], including our single-center experience with non-squamous NSCLC patients [19], Metges et al. reported a survival advantage for NSCLC patients receiving antibiotics prior to the immune-checkpoint inhibitor therapy [55]. In consideration of the limited data and the conflicting results, a prospective evaluation of the impact of antibiotics on clinical outcome with immune-checkpoint inhibition is necessary in future clinical ICB trials.

In our cohort, one out of four patients received denosumab prior to and/or concomitantly with ICB. Anti-RANKL treatment has been shown to protect T-cells specific for melanoma antigens in transgenic mouse models by depletion of medullary thymic epithelial cells, which interferes with a negative selection. Blockade of the RANK-RANKL interaction resulted in improved survival in mice, which had been previously inoculated with melanoma cells [56]. In an animal model, Ahern et al. found that RANK was mainly expressed by tumor-infiltrating lymphocytes while RANKL expression was largely restricted to tumor-infiltrating macrophages, dendritic cells, and myeloid-derived suppressor cells. The co-administration of RANKL-targeting antibodies and immune-checkpoint inhibitors increased intra-tumoral density of CD8^+^ T-cells and enhanced antitumor activity [44,45]. However, denosumab co-administration did not impact the clinical outcome in our analysis (Table 2).

Anti-VEGF targeting therapy in addition to chemotherapy increased the number of CD8^+^ T-cells in the peripheral blood in advanced melanoma patients [57] and improved antigen-specific CD8^+^ T-cell responses in vivo and in vitro in aNSCLC [40]. In animal models, interferon-gamma induced upregulation of PD-L1 drove secondary resistance to VEGF-blockade while concurrent anti-angiogenic therapy and PD-L1 inhibition induced formation of high endothelial venules, which facilitated T-cell infiltration and enhanced anti-tumor activity [58]. Seventeen patients (12%) had been treated with anti-VEGF targeting therapy prior to the initiation of immune-checkpoint inhibition or concomitantly with PD-1/PD-L1 blockade in our NSCLC cohort. Prior or concomitant anti-VEGF blockade was associated with inferior OS (Figure 2B, Table 2B). The latter finding may be explained by immune-checkpoint inhibition being applied in later therapy lines in anti-VEGF exposed patients when compared to anti-VEGF naïve patients (Appendix A). In the IMpower150 study, Socinski et al. investigated carboplatin plus paclitaxel and bevacizumab with and without atezolizumab as front-line systemic therapy in non-squamous aNSCLC and found an OS benefit of 4.5 months with the addition of ICB [12]. However, the report on clinical outcome of patients treated with carboplatin, paclitaxel, and atezolizumab without bevacizumab is eagerly awaited to clarify the role of concomitant anti-VEGF blockade in this setting.

## 5. Conclusions

In conclusion, the results of our analysis demonstrate the practicality of estimating OS from initiating the PD-1/PD-L1 blockade based on baseline ALC and ECOG performance status in aNSCLC. A high baseline ALC was associated with a good ECOG performance status. Stratification according to ALC and ECOG status in future clinical NSCLC trials investigating ICB is warranted. Neither PD-L1 expression status, nor antibiotic treatment status, had an impact on OS in our NSCLC cohort. The question arises whether the negative impact of antibiotic exposure on OS in previous reports is caused by interference with ICB or is caused by the underlying condition necessitating antibiotic administration. The negative impact of anti-VEGF targeting therapy preceding immune-checkpoint inhibitor therapy was unexpected but may be explained by administration of PD-1/PD-L1 inhibitors in later therapy-lines. Further data from the IMpower150 study may eventually clarify the role of anti-VEGF therapy in combination with ICB in aNSCLC.

## Figures and Tables

**Figure 1 jcm-08-01014-f001:**
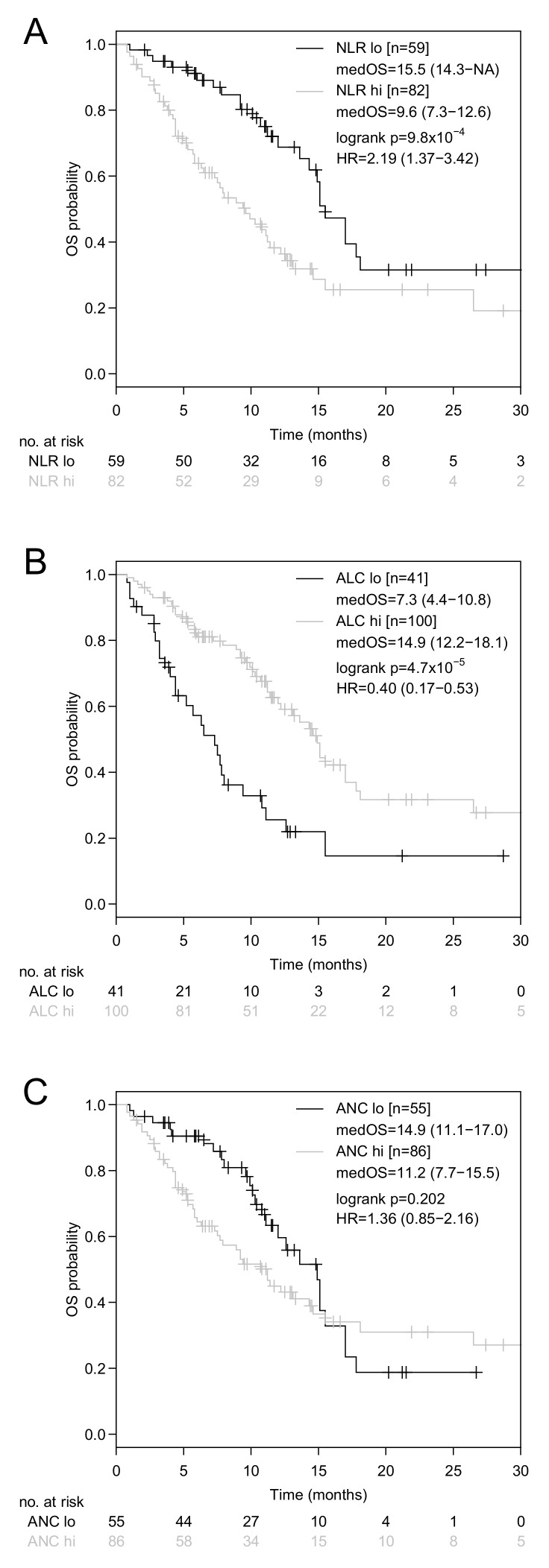
Kaplan-Meier curves for OS according to the baseline neutrophil-lymphocyte ratio (NLR), absolute lymphocyte count (ALC), and absolute neutrophil count (ANC). Comparison of Kaplan-Meier curves for OS in advanced NSCLC patients with a baseline NLR > 3.80 versus ≤ 3.80 (**A**), ALC > 0.93 × 10^9^/L versus ≤ 0.93 × 10^9^/L (**B**), and ANC > 4.83 × 10^9^/L versus ≤ 4.83 × 10^9^/L (**C**). HR is hazard ratio, 95% confidence interval in brackets.

**Figure 2 jcm-08-01014-f002:**
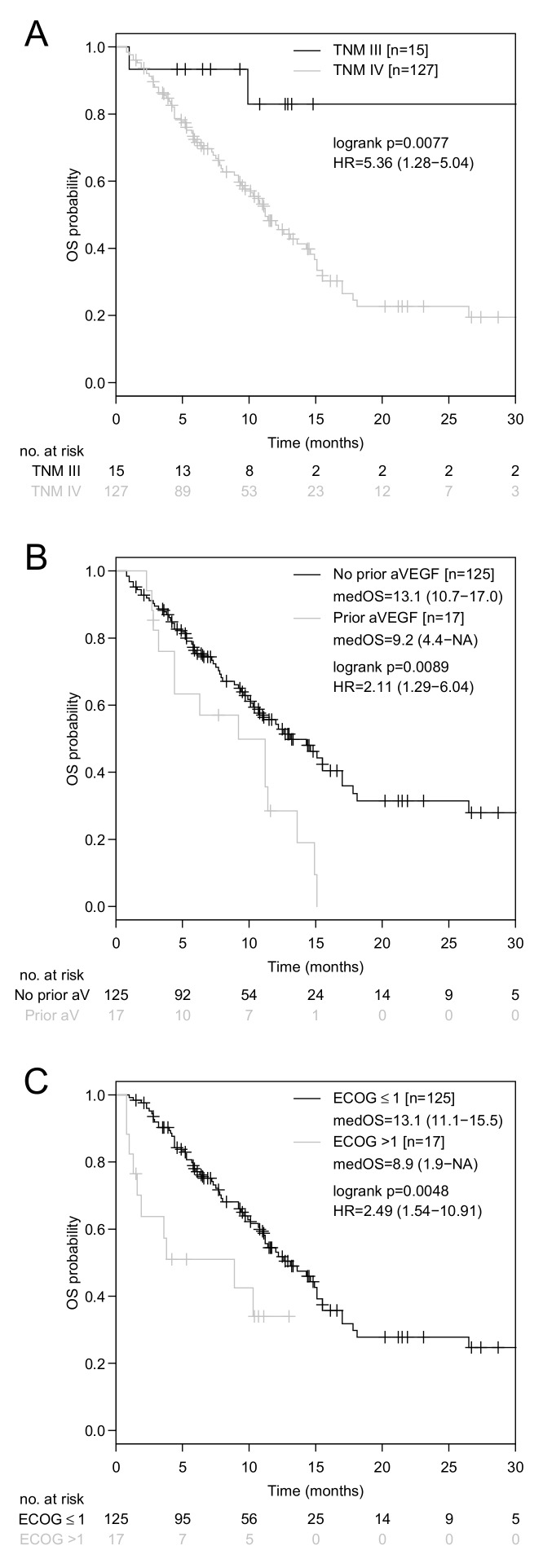
Kaplan-Meier curves for OS according to the TNM stage, prior or concomitant anti-VEGF targeting therapy and ECOG performance status. Comparison of Kaplan-Meier curves for OS in advanced NSCLC patients with TNM stage IV versus III (**A**), administration of prior or concomitant anti-VEGF targeting therapy (yes versus no) (**B**), and ECOG performance status >1 versus ≤1 (**C**). HR is the hazard ratio with a 95% confidence interval in brackets.

**Figure 3 jcm-08-01014-f003:**
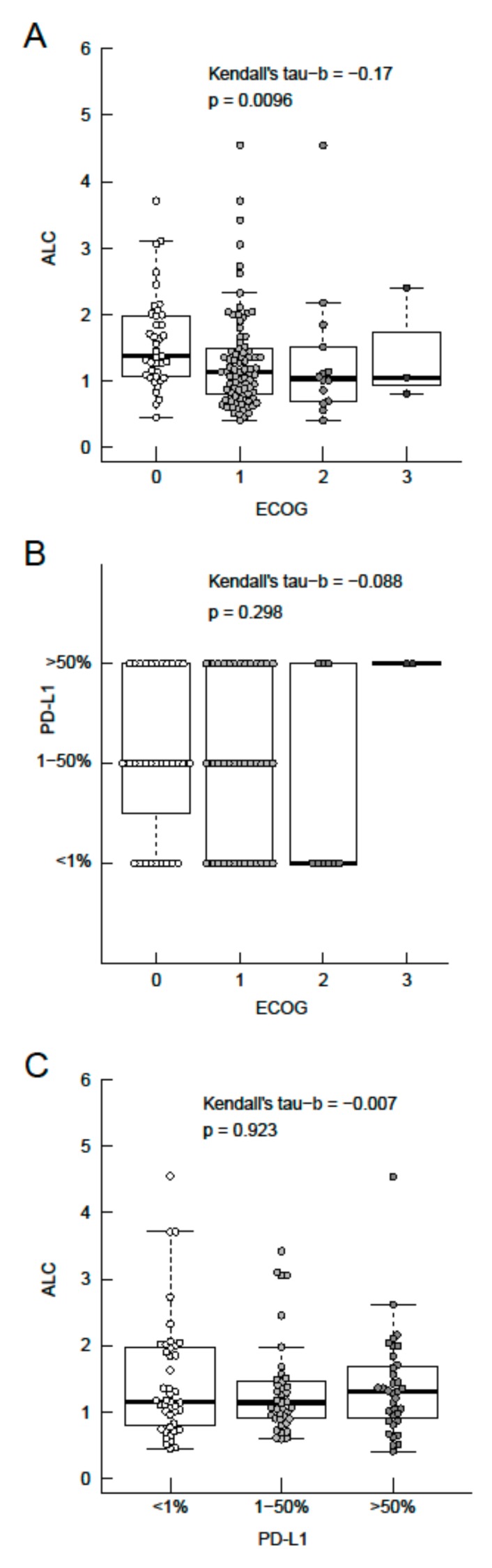
Association between the baseline absolute lymphocyte count (ALC), ECOG performance status, and categorical PD-L1 expression. Association between ALC and ECOG performance status (**A**), categorical PD-L1 expression status and ECOG performance status (**B**), ALC and categorical PD-L1 expression status (**C**). Boxes representing the interquartile range (IQR) from the first quartile (Q1) to the third quartile (Q3) of data, thick line the median, upper whiskers the minimum of (maximum or Q3 + 1.5 × IQR) and lower whiskers the maximum of (minimum or Q1–1.5 × IQR).3.4. Association of Antibiotic Exposure and Survival with Immune-Checkpoint Blockade.

**Table 1 jcm-08-01014-t001:** Baseline characteristics of 142 advanced NSCLC patients.

N = 142 (100%)		
**Age**	Mean (Standard Deviation)	66 (10.6)
**Sex**	male	85 (60%)
	female	57 (40%)
**ECOG performance status**	0	39 (27%)
	1	86 (61%)
	2	14 (10%)
	3	3 (2%)
**Histology**	non-squamous	96 (68%)
	squamous	46 (32%)
**Smoking history**	smoker	116 (88%)
	never-smoker	16 (12%)
	missing	10 (7%)
**TNM stage**	IIIA	6 (4%)
	IIIB	8 (6%)
	IIIC	1 (1%)
	IV	127 (89%)
**ALK translocation**	no	131 (98%)
	yes	3 (2%)
	missing	8 (6%)
**EGFR mutation status**	wild-type	130 (93%)
	mutant	10 (7%)
	missing	2 (1%)
**CNS involvement**	no	112 (79%)
	yes	30 (21%)
**PD-L1 status**	positive	75 (63%)
	negative	44 (37%)
	missing	23 (16%)
**PD-L1 status category**	<1%	44 (37%)
	1–50%	39 (33%)
	>50%	35 (30%)
**ICB therapy line**	1st line	40 (28%)
	2nd line	67 (47%)
	≥ 3rd line	35 (25%)
**Immune-checkpoint inhibitor**	nivolumab	79 (55%)
	pembrolizumab	52 (37%)
	atezolizumab	11 (8%)
**Monotherapy versus combined therapy**	ICB monotherapy	137 (97%)
	ICB combination therapy	5 (3%)
**Tertiary oncologic center**	Salzburg	50 (35%)
	Linz	92 (65%)
**Prior/concomitant denosumab application**	no	106 (75%)
	yes	36 (25%)
**Prior/concomitant anti-VEGF therapy ***	no	125 (88%)
	yes	17 (12%)
**Prior radiotherapy ^#^**	no	79 (56%)
	yes	63 (44%)
**Subsequent therapy**	no therapy	85 (60%)
	taxane-based	19 (13%)
	TKI	17 (12%)
	other	21 (15%)
**Antibiotic treatment during ICB ^§^**	no	80 (56%)
	yes	62 (44%)
**Antibiotic class**	penicillin	45 (73%)
	fluoroquinolone	27 (44%)
	cephalosporine	12 (19%)
	carbapenem	5 (8%)
	metronidazole	5 (8%)
	macrolide	4 (6%)
	linezolide	2 (3%)
**Antibiotic treatment indication**	empiric antibiotic therapy	31 (50%)
	respiratory tract infection	18 (29%)
	perioperative prophylaxis	5 (8%)
	gastrointestinal tract infection	4 (6%)
	biliary tract infection	2 (3%)
	urinary tract infection	1 (2%)
	central venous catheter infection	1 (2%)

ECOG: Eastern Cooperative Oncology Group, EGFR: epidermal growth factor receptor, ALK: anaplastic lymphoma kinase, CNS: central nervous system, PD-L1: programmed cell death ligand 1, ICB: immune-checkpoint blockade, VEGF: vascular endothelial growth factor, TKI: tyrosine kinase inhibitor. * bevacizumab, ramucirumab, or nintedanib. ^§^ administration of antibiotics within a time frame of one month before or one month after initiation of immune-checkpoint blockade. ^#^ to the primary tumor or metastases.

**Table 2 jcm-08-01014-t002:** Univariate and multivariate analysis for PFS (A) and OS (B).

**A**			**Progression-Free Survival**															
			**Univariate Cox Regression Model**								**Multivariate Cox Regression Model**							
		*P* ^$^	N1	N2	N	Events	*P*	HR	Lower 95% CI	Upper 95% CI	N	Events	*P*	HR	Lower 95% CI	Upper 95% CI	*P*	*P*
Antibiotic exposure *	yes versus no	0.080	62	80	142	109	0.922	1.02	0.70	1.49	119	91	-	-	-	-	-	0.047
Neutrophil-lymphocyte-ratio	>3.8 versus ≤3.8	0.202	82	59	141	108	0.097	1.39	0.94	2.04			-	-	-	-	-	-
CNS involvement	yes versus no	0.276	30	112	142	109	0.629	1.12	0.71	1.78			-	-	-	-	-	0.621
C-reactive protein (mg/dL)	>0.6 versus ≤0.6	0.213	94	44	138	107	0.633	1.11	0.73	1.69			-	-	-	-	-	0.139
Tertiary cancer center	Linz versus Salzburg	0.905	92	50	142	109	0.061	0.69	0.47	1.02			-	-	-	-	-	0.798
Histology	squamous versus non-squamous	0.719	46	96	142	109	0.595	0.90	0.60	1.34			-	-	-	-	0.710	0.478
Sex	female versus male	0.509	57	85	142	109	0.257	0.80	0.54	1.18			-	-	-	-	0.209	0.169
TNM stage	stage IV versus stage III	0.541	127	15	142	109	0.044	2.03	1.02	4.03			0.476	1.29	0.64	2.62	0.415	0.519
ICB therapy line	≥3rd line versus <3rd line	0.279	35	107	142	109	0.011	1.73	1.13	2.63			0.610	1.15	0.68	1.94	0.531	0.777
PD-L1 status	PD-L1+ versus PD-L1-	0.212	75	44	119	91	0.001	0.49	0.32	0.75			0.053	0.61	0.37	1.01	0.050	0.105
Age	>66 versus ≤66 years	0.755	67	75	142	109	0.793	1.05	0.72	1.54			-	-	-	-	0.954	0.343
Smoking history	smoker versus never-smoker	0.229	116	16	132	100	0.700	1.14	0.59	2.19			-	-	-	-	-	0.992
Prior/concomitant anti-VEGF therapy	yes versus no	0.530	17	125	142	109	0.029	1.89	1.07	3.36			0.168	1.62	0.81	3.24	0.249	0.063
Prior radiotherapy	yes versus no	0.782	63	79	142	109	0.670	0.92	0.63	1.35			-	-	-	-	-	0.341
Prior/concomitant denosumab therapy	yes versus no	0.316	36	106	142	109	0.190	1.33	0.87	2.03			-	-	-	-	-	0.646
ECOG performance status	>1 versus ≤1	0.927	17	125	142	109	0.001	2.53	1.49	4.29			0.099	1.84	0.89	3.77	0.067	0.141
EGFR mutation status	mutant versus wild-type	0.753	10	130	140	108	0.083	1.84	0.92	3.66			-	-	-	-	-	0.192
ALK translocation	yes versus no	0.239	3	131	134	103	0.467	1.53	0.48	4.86			-	-	-	-	-	0.827
Absolute lymphocyte count (×10^9^/L)	>0.93 versus ≤0.93	0.796	100	41	141	108	0.010	0.58	0.38	0.88			0.012	0.55	0.35	0.88	0.030	0.023
Absolute neutrophil count (×10^9^/L)	>4.83 versus ≤4.83	0.058	86	55	141	108	0.898	1.03	0.69	1.52			-	-	-	-	-	-
**B**			**Overall Survival**															
			**Univariate Cox Regression Model**								**Multivariate Cox Regression Models**							
		*P* ^$^	N1	N2	N	Events	*P*	HR	lower 95% CI	upper 95% CI	N	Events	*P*	HR	Lower 95% CI	Upper 95% CI	*P*	*P*
Antibiotic exposure *	yes versus no	0.117	62	80	142	76	0.675	0.90	0.56	1.45	141	75	-	-	-	-	-	0.135
Neutrophil-lymphocyte-ratio ^$^	>3.8 versus ≤3.8	0.035	82	59	141	75	0.001	2.22	1.36	3.63			-	-	-	-	-	-
CNS involvement	yes versus no	0.499	30	112	142	76	0.154	1.49	0.86	2.57			-	-	-	-	-	0.249
C-reactive protein (mg/dL)	>0.6 versus ≤0.6	0.702	94	44	138	74	0.598	1.15	0.68	1.94			-	-	-	-	-	0.625
Tertiary cancer center	Linz versus Salzburg	0.249	92	50	142	109	0.271	0.77	0.49	1.22			-	-	-	-	-	0.496
Histology	squamous versus non-squamous	0.101	46	96	142	76	0.597	0.87	0.52	1.46			-	-	-	--	0.330	0.413
Sex	female versus male	0.507	57	85	142	76	0.796	0.94	0.59	1.49			-	-	-	-	0.782	0.211
TNM stage	stage IV versus stage III	0.322	127	15	142	76	0.018	5.52	1.35	22.64			0.049	4.18	1.01	17.36	0.040	0.126
ICB therapy line	≥3rd line versus <3rd line	0.930	35	107	142	76	0.004	1.97	1.24	3.16			0.055	1.64	0.99	2.71	0.031	0.271
PD-L1 status	PD-L1+ versus PD-L1-	0.757	75	44	119	61	0.140	0.68	0.41	1.13			-	-	-	-	-	0.878
Age	>66 versus ≤66 years	0.627	67	75	142	76	0.977	0.99	0.63	1.57			-	-	-	-	0.954	0.566
Smoking history	smoker versus never-smoker	0.996	116	16	132	70	0.636	0.84	0.40	1.75			-	-	-	-	-	0.366
Prior/concomitant anti-VEGF therapy	yes versus no	0.176	17	125	142	76	0.010	2.18	1.21	3.93			0.017	2.18	1.15	4.14	0.012	0.002
Prior radiotherapy	yes versus no	0.863	63	79	142	76	0.579	1.14	0.72	1.78			-	-	-	-	-	0.284
Prior/concomitant denosumab therapy	yes versus no	0.847	36	106	142	76	0.849	0.95	0.54	1.65			-	-	-	-	-	0.993
ECOG performance status	>1 versus ≤1	0.159	17	125	142	76	0.007	2.58	1.30	5.10			0.001	3.23	1.58	6.60	0.002	0.887
EGFR mutation status	mutant versus wild-type	0.337	10	130	140	75	0.100	1.93	0.88	4.21			-	-	-	-	-	0.585
ALK translocation	yes versus no	0.365	3	131	134	71	0.255	1.96	0.62	6.27			-	-	-	-	-	0.670
Absolute lymphocyte count (×10^9^/L)	>0.93 versus ≤0.93	0.136	100	41	141	75	<0.001	0.38	0.24	0.62			<0.001	0.38	0.23	0.62	<0.001	0.003
Absolute neutrophil count (×10^9^/L) ^$^	>4.83 versus ≤4.83	0.002	86	55	141	75	0.207	1.37	0.84	2.22			-	-	-	-	-	-

CNS: central nervous system, ICB: immune checkpoint blockade, PD-L1: programmed cell death ligand 1, VEGF: vascular endothelial growth factor, EGFR: epidermal growth factor receptor, ALK: anaplastic lymphoma kinase, HR: hazard ratio, 95%CI: 95% confidence interval. * administration of antibiotics within a time frame of month before to one month after start of immune-checkpoint blockade. ^$^ proportional hazard assumptions are violated, *P*
^$^: *P*-values from test for proportional hazard (cox.zph).

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
