# Peer review of "Baseline Absolute Lymphocyte Count and ECOG Performance Score Are Associated with Survival in Advanced Non-Small Cell Lung Cancer Undergoing PD-1/PD-L1 Blockade"

_jcm, 2019, doi:10.3390/jcm8071014_

Reviewer 1 Report

Review of Baseline absolute lymphocyte count and ECOG performance score as predictors of survival in advanced non-small cell lung cancer undergoing PD-1/PD-L1 blockade

General comments

This analysis examined the baseline characteristics of patients with advanced NSCLC undergoing PD-1/PD-L1 therapy and the association with survival outcomes. Although the title of the paper is “Baseline absolute lymphocyte count and ECOG performance score as predictors of survival in advanced non-small cell lung cancer undergoing PD-1/PD-L1 blockade”, this is not a predictive modeling exercise and no validation of the predicting power of ALC and ECOG score on survival outcomes was carried out. Only simple regressions were run to test the association between survival outcomes and ALC/ECOG and the model choice was not sufficiently justified. Therefore, I think the title is a bit misleading. 

However, the data that the authors collected contained very detailed lab parameters and tumor characteristics of this patient population. This paper is more of a good descriptive analysis of patients receiving PD-1/PD-L1 treatments in the real world than predicting survival outcomes of aNSCLC patients using several parameters. 

Please see my specific comments below.

Specific comments

1. The last statement of section “Patients” is confusing to me (line 120-122). Please be more clear about how PD-1/PD-L1 inhibitors were prescribed in the patient population (e.g. specify what international standards were followed, which clinical trials and what early access programs you are referring to here).

2.  In your study, PD-L1 positivity was defined as “>=1% PD-L1 positive tumor cells”. As you have already pointed out in the text, this is an arbitrary cut-off. Did you test the impact of using different cut-offs for PD-L1 positivity on the results? Or perhaps consider adding the justification of choosing >=1% as the cut-off value. 

3. The definitions of PFS and OS do not seem right to me. First, those who are lost to follow up should be censored in the analysis, not considered the same as those who died. Second, if PFS is also defined as start of ICB until death, how do you distinguish PFS from OS in your analysis?

4. Why patients are grouped into anti-VEGF therapy exposed and anti-VEGF naïve were not explicitly explained (in your baseline characteristics tests and Table S1 results). 

5. In the Statistical Analyses section, why only including significant variable in the univariate cox regression into multivariate cox regression model? The established risk factors and clinically relevant variables should all be included regardless of statistical significance (e.g. you omitted age and sex in your multivariate model). In addition, since you are grouping patients into two groups, are you running separate models for these two groups? 

6. It seems redundant to describe Table 1 in words. 

7. In Results section, the figures presented are not clear if they are raw survival curves or adjusted survival curves. If it’s KM curves, then it’s unadjusted, so the figures should not be associated with the results from multivariate models. If it's adjusted curves, then please specify that in the text. 

8. What is the implication for the data presented in Table S1? Shouldn’t this be done prior to running regressions? 

9. The conclusions on association based on results from only using a Kendall’s rank correlation coefficient seems very weak. There are many variables in your data that could and should be controlled for. 

10. “Maximal Harrell’s C-index” was mentioned in Statistical Analyses section once and no results were reported on that at all. What is the “the optimal cut-off value for OS prediction for continuous data such as the ALC, ANC and NLR”? In addition, not all tests mentioned in your statistical analysis section were subsequently discussed in the results (e.g. what is the result of testing proportional hazard assumption?). 

11. In the Conclusion section, the rationale of including several variables were explained, which I think might be better to put into intro or methods so the readers without a clinical background could understand why these variables are important to explore.

Reviewer 2 Report

In this manuscript, Huemer investigated the correlation between baseline characteristics and the therapeutic efficacy of PD-1/PD-L1 inhibitors in 142 NSCLC patients. Predicting the effect of immune checkpoint therapies is a challenging topic in oncology, this work has great significance.

1. In table 1, for "Age", it is better to give mean and standard deviation.

2. In Figure S1, please label A and B. Also put numbers at risk in this figure.

3. Model selection, such as stepwise selection, is a better way to select significant variable of interest into the model rather than selecting variables one by one through univariate cox regression analysis.

4. Since the sample size is limited and it has so many variables of interest to be included, propensity score method would be a better method when constructing cox regression model.

5. In order to find the cut-off value, ROC curves are generally used. However, the author used the C-index to analyze the cut-off value (line 154). Could the author explain the difference between the two? Or, compared with the ROC curve, what is the advantage of calculating C-index?
